# Design, Formulation, and Characterization of Valsartan Nanoethosomes for Improving Their Bioavailability

**DOI:** 10.3390/pharmaceutics14112268

**Published:** 2022-10-24

**Authors:** Ali M. Nasr, Fayrouz Moftah, Mohammed A. S. Abourehab, Shadeed Gad

**Affiliations:** 1Department of Pharmaceutics, Faculty of Pharmacy, Port Said University, Port Said 42526, Egypt; 2Department of Pharmaceutics and Industrial Pharmacy, Faculty of Pharmacy, Galala University, New Galala 43713, Egypt; 3Department of Pharmaceutics, Faculty of Pharmacy, Sinai University, Arish 45511, Egypt; 4Department of Pharmaceutics, Faculty of Pharmacy, Umm Al-Qura University, Makkah 21955, Saudi Arabia; 5Department of Pharmaceutics, College of Pharmacy, Minia University, Minia 61519, Egypt; 6Department of Pharmaceutics, Faculty of Pharmacy, Suez Canal University, Ismailia 41522, Egypt

**Keywords:** antihypertensive drug, valsartan, ethosomes, transdermal drug delivery system, bioavailability

## Abstract

The objective of this study was to formulate and evaluate valsartan (VLT) ethosomes to prepare an optimized formula of VLT-entrapped ethosomes that could be incorporated into a sustained release transdermal gel dosage form. The formulation of the prepared ethosomal gel was investigated and subjected to in vitro drug release studies, ex vivo test, and in vivo studies to assess the effectiveness of ethosomal formulation in enhancing the bioavailability of VLT as a poorly soluble drug and in controlling its release from the transdermal gel dosage form. The acquired results are as follows: Dependent responses were particle size, polydispersity index, zeta potential, and entrapment efficiency. The optimized VLT-ETHs had a nanometric diameter (45.8 ± 0.5 nm), a negative surface charge (−51.4 ± 6.3 mV), and a high drug encapsulation (94.24 ± 0.2). The prepared VLT ethosomal gel (VLT-ethogel) showed a high peak plasma concentration and enhanced bioavailability in rats compared with the oral solution of valsartan presented in the higher AUC (0–∞). The AUC (0–∞) with oral treatment was 7.0 ± 2.94 (μg.h/mL), but the AUC (0–∞) with topical application of the VAL nanoethosomal gel was 137.2 ± 49.88 (μg.h/mL), providing the sustained release pattern of VLT from the tested ethosomal gel.

## 1. Introduction

The transdermal drug delivery system (TDDS) is regarded as an ideal route of medication administration for drugs with a restricted bioavailability. When compared to parenteral and oral routes of administration, drug administration by the transdermal route has the potential to improve patient compliance with dose control while also resolving issues such as enzymatic drug degradation, hepatic first-pass metabolism, and pain from intravenous administration. Transdermal systems can be designed to promote skin penetration and allow for proper controlled absorption of the drug into the bloodstream in varied forms [1,2].

The efficiency of transdermal delivery systems has been widely studied over the past three decades as vesicles carriers, for example, liposomes [3], niosomes [4], transfersomes, and ethosomes [5]. Classic liposomal systems tend to form a drug reservoir in the stratum corneum’s upper layers and prevent drugs from penetrating deeper layers of the skin because of their diminished transdermal delivery of the drug. In comparison to conventional liposomes, ethosomes have been shown to pass the corneal barrier and exhibit significantly better transdermal permeability than liposomes.

The non-invasive nanoparticulate delivery systems, such as ethosomes, can improve drug delivery through the skin. Phospholipids, ethanol at high concentrations, and glycols are all included in these vesicular carriers. Drug penetration via the skin lipids could be enhanced by the synergistic impact of ethanol in ethosome formulations. In addition, the soft flexible vesicles can pass deep into the fractured skin bilayers due to the ethanol effect, which enhances the membrane permeability as well as the fluidity of lipid molecules [6]. This class of biodegradable ethosomes contains high concentrations of alcohol, which has resulted in the formation of a negative charge, resulting in increased drug delivery and bioavailability [7]. Ethosomes have the ability to encapsulate and distribute hydrophilic and lipophilic medications through skin layers, owing to their particular structure and great lamellarity. Buspirone [8] and betahistine dihydrochloride [9] are examples of hydrophilic drugs, while lipophilic drugs include simvastatin [10], valsartan [11], and triamcinolone acetonide [12].

Additionally, vesicular systems can also be added to hydrogel bases in order to increase formulation’s delivery and improve its stability [13]. Carbomers and cellulose polymers, either natural or synthetic, are the main components of hydrogels. In addition to having sufficient bioadhesive properties and viscosity, these polymers have also been revealed to be compatible with nanoethosomes and to improve drug solubility and delivery [14].

Valsartan (VLT) is an orally active drug. It is used to treat high blood pressure. It is a highly selective angiotensin II type 1 receptor antagonist. First-pass effect and poor absorption from GIT were observed. A decrease in Cmax and AUC of VLT due to food intake is around 50% and 40%, respectively, which may lead to diminished pharmacological effects [15]. As a result, its oral bioavailability (BA) ranges between 10% and 35%. The transdermal drug delivery system (TDDS) can help hypertensive patients overcome these VLT drawbacks and provide even more therapeutic benefits [16]. For transdermal dosage forms, VLT possesses the ideal biological properties, including limited oral bioavailability (10–35%), elimination half-life (4–6 h), small molecular weight (435.5 g/mol), and increased lipid solubility (Log P = 1.499).

As a result, developing sustained transdermal drug delivery systems for long-term hypertension treatment necessitates a successful approach. The main objective of this research was to design and develop lipid-based nanoethosomes in a gel dosage form containing VLT in order to improve epidermal permeability and pharmacodynamic efficacy while avoiding the limitations of oral administration.

## 2. Materials and Methods

### 2.1. Materials

#### Chemicals

Valsartan was kindly provided by the Egyptian International Pharmaceutical Industries Company (EIPICO) (Cairo, Egypt). Phospholipon^®^ 90 G was provided as a promotional sample by Lipoid GmbH (Nattermannallee, Köln, Germany). Pure soybean lecithin was bought from the DASIT Company (Val de Reuil, France). Ethanol was bought from El-Nasr Pharmaceutical Chemicals Co., ADWIC (Abu Zaabal, Egypt). Transcutol P, a penetration enhancer, was a kind gift sample from Gattefossé (Saint-Priest, France), and propylene glycol, a penetration enhancer, was provided from ADWIC, El-Nasr Pharmaceutical Chemicals Co. (Abu Zaabal, Egypt). The hydrogel base was provided by (EIPICO) Egyptian International Pharmaceutical Industries Company (Cairo, Egypt). Acetonitrile and methanol (HPLC grade) were purchased from the Sigma Chemical Company (St. Louis, MO, USA). All supplementary chemicals were of analytical grade. No further purification was carried out.

### 2.2. Preparation of VLT-Loaded Ethosomes (VLT-ETHs)

VLT-ETHs were prepared by the “Hot” method, with the proper modifications as described in (United States Patent –5, 540, 934) [17], followed by sonication. The first stage involved creating a colloidal solution by heating the VLT (80 mg drug) and phospholipid (Phospholipon 90 G or soya lecithin) in a water bath at 40 °C. In a separate container, ethanol and a penetration enhancer (transcutol P or propylene glycol) were combined and heated to 40 °C. The organic phase was added to the aqueous phase once both mixtures had achieved 40 °C while the mixture was continuously agitated on a magnetic stirrer for 10 min (Brandstead/Thermolyne, Swedesboro, NJ, USA) [18,19]. Finally, in order to reduce the size of the ethosomal formulation’s vesicles to the required level, for 5–10 min at 60 rpm, all formulations were subjected to a digital sonifier (Branson, Danbury, CT, USA) [20]. For further investigation, all formulations were preserved in tightly sealed containers and stored in a refrigerator at 4 °C.

### 2.3. Preparation of VLT-Loaded Ethogel

The Carbopol 940 (Carb.940) (0.15% *w*/*w*), a gelling agent, was used in the preparation of the VLT nanoethsomal gel (VLT-ethogel). It was necessary to gradually dissolve the accurate weight of the polymer in 60 mL boiling distilled water while stirring with a magnetic stirrer at 800 rpm. Homogeneous and smooth gels were then developed without any lumps. Adding triethanolamine dropwise to the carb.940 hydrogel resulted in a homogeneous semisolid gel with a pH of 5.5. For ideal gel dispersion, the obtained gel was kept overnight in a refrigerator at 4 °C [21]. In order to ensure that the VLT nanoethosome formulation was distributed evenly throughout the gel base, it was necessary to mechanically mix the VLT nanoethsomal formulation pellets with the hydrogel base on a magnetic stirrer (Brandstead/Thermolyne, Swedesboro, NJ, USA) at 200 rpm for 10 min [22]. Distilled water was used to bring the final weight up to 100 g.

### 2.4. Screening Formulations for Build Information

Initial screening was carried out to determine the most essential factors and the most important excipients that might impact the preparation of VLT-ETHs. The five factors were selected with varying levels for a total of 16 formulations. Dependent responses were the mean particle size (PS), zeta potential (ZP), entrapment efficiency (***EE***%), and polydispersity index (PDI). Table 1 provides a summary of the factors studied and their levels.

### 2.5. Optimization and Selection of Most Significant Factors

Optimization and selection of factors and excipients was applied to select the most significant factors obtained from the prior screening. This resulted in multiple formulations with the best particle size, zeta potential, entrapment efficiency, and polydispersity index. Three factors were selected with varying levels for a total of 18 formulations. The three selected factors were A: ethanol concentration; B: lipid concentration, and C: sonication time. Dependent responses were particle size (Y1), zeta potential (Y2), entrapment efficiency (***EE***%) (Y3), and polydispersity index (PDI) (Y4). The objectives were to minimize the particle size (Y1) and polydispersity index (PDI) (Y4), to obtain a zeta potential value (Y2) from −25 mV to −45 mV, and to maximize the entrapment efficiency (***EE***%) (Y3).

### 2.6. Particle Size (PS) and Zeta Potential (ZP)

Particle size (PS), zeta potential (ZP), and polydispersity index (PDI) of the prepared VLT-ETH formulations were determined using the dynamic light scattering (DLS) technique. Using a computerized inspection system (Zetasizer Malvern Instruments Ltd., Malvern, UK) at 25 °C and a scattering angle of 90°, these responses were accurately measured. Before analysis, each sample (0.5 mL) was diluted 10-fold in distilled water and thoroughly mixed to prevent multi-scattering action. For this, a clear disposable zeta cell was filled with diluted samples to determine the zeta potential (ZP) [23,24]. All measurements were made three times, and the average and standard deviation were calculated (SD).

### 2.7. Entrapment Efficiency (**EE**%)

A predetermined volume of freshly prepared VLT-ETH vesicles was centrifuged in a high-speed cooling centrifuge [25]. Every ethosomal sample (1 mL) was ultracentrifuged for 1 h at 4 °C at 15,000 rpm using a cooling centrifuge (2-16KL, Sigma Laborzentrifugen GmbH, Osterode am Harz, Germany) [26,27]. The sediment and supernatant liquids were separated, and the free drug (unentrapped) concentration was determined in the supernatant spectrophotometrically UV spectrophotometer (Tokyo, Japan). The absorbance of the drug was noted at 250 nm. Valsartan entrapment percentage was calculated as the ratio of the detected entrapped drug amount to total drug amount contained in each formulation [28,29]. According to the following equation NO.1:(1)$$EE\%=\frac{\left(Qt-Qs\right)}{\left(Qt\right)}\times 100$$
where EE% is the entrapment efficiency percentage, Qt is the theoretical VLT amount that was actually added in each formulation, and Qs is the VLT amount that was identified only in the supernatant.

### 2.8. In Vitro Drug Release

The in vitro release of VLT was determined from the prepared nanoethosomal formulations (R1–R18), optimized formulation (OPT-VLT), and ethogel formulation (VLT-Ethogel). The dialysis bag technique was utilized to perform this test in order to determine the pattern of in vitro drug release. This test was performed by cellophane membrane dialysis tubing with a molecular weight cut-off of 12,000–14,000 Da (SERVAPOR^®^ Dialysis Membranes, Heidelberg, Germany). To ensure complete wetting of the membranes, they were already soaked overnight prior to dialysis in the receptor medium (phosphate buffer, pH 7.4) [30]. This study was performed by a USP dissolution tester (Apparatus I). Five milliliters of ethosomal suspension were placed in cylindrical tubes (2.5 cm in diameter and 6 cm in length). Each tube was entirely covered by a molecular porous membrane on one end and attached to the shafts of the apparatus on the other end, instead of baskets. The shafts were then lowered into vessels containing 250 mL of PBS (pH 7.4) at a temperature of 37 ± 0.5 °C and a rotational speed of 100 rpm. To maintain a constant volume, aliquots of dissolution medium (5 mL) were withdrawn at specified time intervals of 0.5, 1, 2, 3, 4, 5, 6, 8, 12, 24, and 48 and replaced with fresh medium.

Filtration with a 0.22 µm nylon syringe filter and spectrophotometric analysis at 250 nm were performed on the samples. Triplicate trials were conducted against a blank. To investigate the VLT release kinetics model, the collected data were subjected to kinetic treatment using the zero, first, and Higuchi diffusion models, as well as the Hixson–Crowell model [31]. In each case, the correlation coefficient (r) was calculated.

### 2.9. Ex Vivo Permeation Study through Rat Abdominal Skin

Using Franz’s diffusion cell (Maharashtra, Mumbai, India), an ex vivo skin permeation study was performed on rat skin [32]. Wistar rats (male albino) were needed to use their abdomen skin. They were 6–8 weeks old and weighed 120–150 g. To avoid ethosomal gel adsorption to the hair, the hair was removed. The abdomen skin was properly checked and detached from any adhesive subcutaneous tissue and/or fat with a scalpel to avoid the presence of surface defects such as crevices or fine holes in the regions. After washing the prepared skin with double-distilled water, it was wrapped in aluminum foil and stored at −18 °C to prolong its metabolic efficiency.

Prior to the permeation study, the skin was hydrated in phosphate buffer (pH 7.4) at a temperature of 25 °C overnight to eliminate foreign matter and leachable enzymes [33,34]. Between the donor and receptor compartments of the apparatus, the shaved rat skin (available surface area = 1.7662 cm^2^) was placed. Thus, the stratum corneum was exposed to the donor compartment, whereas the dermis was exposed to the receptor compartment. PBS (pH 7.4; diffusion medium = 7 mL) was filled in the receiver compartment and kept at 37 ± 1 °C. Once the membrane came in contact with the surface of the receptor media, a small magnetic stirrer was used to constantly agitate the diffusion medium at a speed of 100 rpm. A total of 1 mL of VLT ethosomal gel (VLT-ethogel) was added to the dorsal side of the rat skin in the donor compartment and then covered with aluminum foil to prevent evaporation. At specified sampling points (0.5, 1, 2, 3, 4, 6, 8, 12, 24, 48, and 72 h), a 0.5 mL medium sample was taken and immediately replaced with another 0.5 mL of appropriately warmed buffer in order to maintain the sink condition for 72 h. To avoid interference, these samples were filtered using a 0.22 µm nylon syringe filter and evaluated at 250 nm using a UV spectrophotometer (Tokyo, Japan) against the receptor medium as a blank. The experiment was performed in triplicate with calculation of means and standard deviations.

The cumulative amounts of drug infused through the skin were graphically plotted against time (t) for VLT-ethogel. Additionally, the permeation results of VLT-ethogel were kinetically analyzed in order to determine the order of drug permeation. The steady state flux (***JSS***) was calculated from the slope of the linear part of the cumulative amount of VLT permeated per unit area (µg/cm^2^) against a time (h). The permeability coefficient (***KP***) of VLT across the skin of rats was determined using the Equation (2):(2)$$KP=\frac{JSS}{Co}$$
where ***KP*** is the permeability coefficient, ***JSS*** is the steady state flux, and ***Co*** is the initial VLT concentration.

### 2.10. Differential Scanning Calorimetry (DSC)

The thermal analysis of lyophilized chosen VLT-ETH (optimized formulation), pure powder of VLT, and physical mixture (drug and lipid) was performed using differential scanning calorimetry (SETARAM Inc., Provence-Alpes-Cote d’Azur, France). The physical mixture included the same amount of VLT as the optimized formulation of VLT-ETH. Calibration of the instrument was performed by the standards (mercury, indium, tin, lead, zinc, and aluminum). Purifying gases such as nitrogen and helium were utilized. Small amounts of samples (3–5 mg) were carefully weighed and put into small aluminum pans. The filled pans were press-sealed and covered with aluminum foil. Empty pans served as a reference. We maintained a nitrogen flow rate of 30 mL/min while heating a zone between −20 °C and 237 °C at a rate of 10 °C/min. All of the samples were weighed in an aluminum crucible with a capacity of 120 µL and a thickness of 0.1 mm, and then they were subjected to DSC analysis. Data processing software (CALISTO Data processing software version 149) was used to process the thermogram results [35].

### 2.11. Fourier Transform Infrared Spectroscopy (FTIR)

FTIR spectroscopy was used to conduct a drug–carrier interaction study to determine the compatibility of the drug with the ethosomal components. The FTIR spectra of an improved ethosomal formulation, a pure VLT powder, and a physical mixture were compared [36]. The potassium bromide press technique was used to generate FTIR spectra. A Fourier transform infrared spectrophotometer (FTIR-4100 Jasco, Tokyo, Japan) was used to analyze the samples. A total of 3–5 mg of samples were blended with dry infrared crystalline potassium bromide (IR-grade) compacted into discs under vacuum. The recorded spectra had a resolution of 0.48–1.93 cm^−1^ and a range of 400–4000 cm^−1^ [37].

### 2.12. Transmission Electron Microscopy (TEM)

Transmission electron microscopy (TEM) was used to easily characterize and visualize the appearance and morphology of the valsartan nanoethosome vesicles. To summarize, one drop of a diluted optimized nanoethosomal mixture was immersed into a carbon-coated grid in. The drop was then placed on the carbon-coated grid for two minutes to allow the ethosomes to dry and adhere. The morphology of the vesicles was investigated using a transmission electron microscope (JTEM model 1010, JEOL^®^, Tokyo, Japan) operating at a 100 kV accelerating voltage [38,39].

### 2.13. In Vivo Study

#### 2.13.1. Experimental Animals

Wistar albino rats (either sex) (6–8 weeks/100–200 g) were obtained from the Animal House at Suez Canal University’s Faculty of Medicine; they were used for in vivo ethosome evaluation. Rats were trained to adhere to experimental humidity and temperature conditions one week before the experiment. Rats were fed rat pellets on a regular basis and were allowed free access to water. During the experiment, rats were housed in a standard laboratory setting with a 12 h light/dark cycle at 25 ± 2 °C. The research technique was verified and approved by the Institutional Animal Ethics Committee of Suez Canal University.

#### 2.13.2. Pharmacokinetic Study

Bioavailability experiments have been conducted using Wistar albino rats as animal models. The animals were chosen after a superficial examination of the skin surface for abnormalities. Reservoir-type transdermal therapeutic systems (TTS) were tested in in vivo pharmacokinetic studies using the optimized valsartan ethosomal gel formulation (VLT-Ethogel-TTS). For the investigation, only rats weighing between 100 and 200 g were selected. In this procedure, approximately 10 cm^2^ of rodent skin was shaved on the abdomen. Rats were maintained under observation and were starved for 24 h prior to TTS application in order to assess any deleterious effects of shaving. On the basis of the weight of the rat, the rat dose (3.6 mg/kg) was determined using the surface area ratio [40,41].

The rats were placed into three groups (*n* = 4) for the experiment.

Group A was considered to be a healthy group and was administered water only.

Group B was given VLT solution orally using the oral feeding device.

Group C received optimized formulation of VLT ethosomal gel–TTS (VLT-Ethogel-TTS).

Throughout this test, blood samples were taken at various times (0, 2, 4, 8, 16, 24, and 48 h). Diethyl ether was used to anaesthetize the rats, and blood samples (1500 µL) were taken from the rat tail vein and placed in Eppendorf tubes containing 8 mg of disodium EDTA as an anticoagulant. Collecting blood samples was followed by careful mixing with the anticoagulant, followed by a brief vortex mixing, and then the mixture was centrifuged for 5 min at 10,000 rpm in a high-speed centrifuge cooled to 4 °C and kept at −80 °C before being subjected to drug analysis using a specified HPLC system with minor modifications [42].

##### Chromatographic Conditions

The mobile phase consisted of a mixture of 20 mM of phosphate buffer (pH 2.68)/cetonitrile in the ratio of 60:40%. The mode of elution was isocratic. The flow rate was 1 mL/min. The column was Column Xterra Ms C18 4.6 × 100 mm. The determination was performed at a 225 nm wavelength. The retention time of VLT was 6.1 min.

##### Preparation of Standard Stock Solution of VLT in Methanol

A total of 10 mg valsartan was weighed in 10 mL methanol (stock solution). A total of 100 µL was taken from stock solution in 100 mL mobile phase to provide a concentration of 1 µg/mL, diluted to produce concentrations of 100 ng/mL in 20 mL mobile phase, then filtered using a 0.22 µm syringe filter, and following this, 100 µL was injected. These solutions were kept in a refrigerator at 4 °C.

##### Sample Preparation

A total of 50 µL from blank plasma was added to 100 µL orthophosphoric acid and 600 µL methanol. The mixture was extracted using vortex for 1 min and centrifuged at 10,000 rpm for 5 min. The supernatant was transferred to a 10 mL volumetric flask, and 1 mL of mobile phase was added; then, the mixture was filtered using a 0.22 µm syringe filter.

After this, 100 µL was injected, and the procedure was repeated with the addition of the standard of valsartan at 100 ng/mL.

##### Preparation of VLT Oral Solution

The surface area ratio was used to calculate the rat dose (3.6 mg/kg) on the basis of the rat’s weight [40,41]. Weights of Wistar albino rats ranged from 150 to 200 gm. Therefore, the calculated dose for each rat was 0.72 mg. The oral VLT solution was produced by dissolving 2.88 mg of VLT in 4 mL of distilled water. One milliliter of the prepared oral solution was provided to each rat.

## 3. Results and Discussion

### 3.1. Screening Formulations for Build Information

An adequate screening study containing the most relevant factors for VAL-ETH preparations was developed using a multi-factor screening approach. A five-factor screening study with more than one level was conducted in order to research a large number of factors at a low cost and with limited resources, as well as to manage varied levels for each of the factors studied (Table 1). Ethosomes were investigated as a carrier for the antihypertensive drug VLT in this study. To this aim, several formulations were prepared with different ethanol and lipid concentrations and sonication times in order to identify the factors that affect PS, ZP, PDI, and ***EE***%. The formulation parameters, as well as their measured responses for the 16 runs, are summarized in Table 2. Table 3 summarizes the observed response coefficients of the formulations, as well as the ANOVA findings generated for measured responses, such as the *p*-value. The sequential model recommended for analyzing the different parameters was linear, according to statistical analysis using ANOVA. As shown in Table 2, the formulations containing 1–4% *w*/*v* PL, 10–40% *v*/*v* ethanol, and 10% *v*/*v* propylene glycol/Transcutol P with different sonication times had a white colloidal appearance with no precipitation of free drug, according to the results. This could have been owing to the synergistic effect of ethanol and penetration enhancer as a co-solvent system with micelle solubilization of PL, greatly increasing valsartan solubility in the formulation. Valsartan-loaded ethosomes were unable to be conducted once the ethanol concentration was increased to 45% *v*/*v* because lipid vesicle rupture occurred. The effect of high ethanol concentration on the lipid vesicles could describe the phase separation of the ethosomal formulations. The significant increase in ethanol concentration (45% *v*/*v*) almost definitely caused the vesicle membrane to leak, resulting in a fall in ***EE***% and, finally, free drug precipitation. As the hydration medium containing 25–40% *v*/*v* ethanol provided the best physical characteristics, formulations containing an ethanol concentration around 25% were selected for further optimization. Similar findings were obtained by Verma and Fahr, Limsuwan and Amnuaikit, and Jin-guang et al. [43,44,45].

#### 3.1.1. Effect of the Independent Factors on Particle Size (PS)

The size of the vesicles in transdermal drug delivery systems must be precisely measured. VLT-ETHs with lower PS values are better formulated.

Nava et al. [30] reported the ideal vesicle size of ethosomes for drug administration to the skin as up to 300 nm in order to ensure that the drug is able to penetrate deeper into the skin and that greater skin permeation is possible. Verma and du Plessis et al. [46,47] reported that small-sized vesicles deliver their contents more efficiently to deeper layers of skin in topical drug delivery systems. For each of the freshly prepared ethosomal formulations, the DLS technique by a computerized Zetasizer was used to determine the vesicle size values (F1–F16), and the results are recorded in Table 2. The vesicles of the developed VLT-ETH formulations ranged from 133.80 ± 4.10 (F5) to 1855.33 ± 4.00 nm (F10), depending on the effect of excipient concentration on the nano-size of the vesicles.

Vesicles with a diameter greater than 600 nm adhere to the stratum corneum and may dry to form a lipid layer on the skin [48,49]. Vesicles relatively smaller than 300 nm were capable of delivering their drugs into deeper layers of skin to a certain extent, whereas the highest drug delivery needed particles smaller than 70 nm [46].

In the statistical analysis of the study coefficients, the significance of the factors and their effect on different dependent variables were demonstrated. A rise in the response was indicated by a positive sign, while a lowering in the response was indicated by a negative sign (decrease). The particle size was significantly influenced by lipid concentration (A) and lipid type (D). Sonication time and penetration enhancer type had no significant impact on PS, as shown in Table 3. According to Anita et al., the size of the vesicles did not change significantly (*p* > 0.05) when the ethanol concentration was changed [50].

##### Effect of Lipid Concentration (A) on Particle Size (PS)

One of the two factors that had a significant impact on particle size was lipid concentration (A). It was found that lipid concentration (A) had synergistic effects on PS, as evidenced by the statistical interpretation of the coefficients (Table 3). Vesicles in Table 2 ranged in size from 187.37 ± 6.30 nm to 635.83 ± 3.00 nm for formulations with concentrations of 1% lipid, while formulations with concentrations of lipid (4%) ranged in size from 1723.00 ± 18.00 nm to 1855.3 ± 4.00 nm. As a result, increasing concentrations of PL (1–4% *w*/*v*) led to significant increases (*p* < 0.05) in the vesicles’ size. The decrease in lipid concentration resulted in decreased PS at 187.37 nm, confirming their suitability for drug delivery across the skin. With increasing lipid concentrations, the PS of ethosomal vesicles increased. Pathan et al. [51] found that increasing the concentration of soya lecithin resulted in an increase in mean particle size. With a formulation containing 1% soya lecithin, small vesicles were formed. The size of ethosomes doubled when the concentration of soya lecithin was increased twofold. When PL concentration was increased, the vesicles grew larger, which was likely due to the PL stiffening them and causing the formation of a thicker matrix structure. Garg et al., Limsuwan et al., and Yang et al. reported similar results [23,52,53]. The sonication time (5–10 min), as shown by the VLT-ETH formulations, clearly had no effect on the vesicle size values.

##### Effect of Type of Lipid (D) on Particle Size (PS)

This system’s formulation has included phospholipids from various sources. The selection of phospholipid type and concentration for the formulation were important factors during the development of the ethosomal system, as the ethosomal system’s size, entrapment efficiency, Z-potential, stability, and penetration properties would be all affected by the formulation’s choice of phospholipid type and concentration. Ethosomal systems have been prepared using two different types of lipids, which are listed in Table 2.

It was one of two variables that had a significant impact on particle size: lipid type (D). The lipid type (D) had an antagonistic impact on PS (decreased), according to the statistical interpretation of the coefficients (Table 3). The formulations with lecithin had the size of the vesicles in a range of 133.80 ± 4.10–635.83 ± 3.00 nm, while in formulations with Phospholipone 90 G, the PS of vesicles ranged from 671.20 ± 2.80 to 1079.67 ± 46.30 nm. As a result, the incorporation of lecithin reduced the vesicle size significantly (*p* < 0.05). Trivedi et al. also confirmed these findings. There was less vesicular formation and less or no aggregation in liposomes prepared with different soy lecithin concentrations. It is possible that the utilization of soya lecithin reduced the vesicle size because the number of phospholipids contained in the lipid bilayer decreased [54]. Thus, a concentration of around 1% lecithin would be ideal as a lipid type for the next optimization study.

#### 3.1.2. Effect of the Independent Factors on the Zeta Potential (ZP)

An important factor in both the stability of the vesicles and skin–vesicle interactions is the charge of the ethosomal vesicles. It is possible to estimate the colloidal system’s stability on the basis of the magnitude of its zeta potential. Ethosomal formulation was tested for stability using zeta potential measurements. There will be no tendency for the particles to come together if all of the suspension particles have a large negative or positive zeta potential, so they will repel each other. Even if the zeta potential values of the particles are low, there will be a force to prevent the particles from coming together and flocculating [55]. In order to stabilize electrostatics in dispersed systems, a ZP of between −35 and −25 mV is required [56].

Table 2 shows that the zeta potentials of the various formulations from F1 to F16 ranged from −1.10 ± 1.40 to 25.73 ± 0.40 mV, which indicates that the formulations did not aggregate quickly [55]. F5 had the lowest zeta potential (−1.10 ± 1.40 mV), while F7 had the highest (− 25.73 ± 0.40 mV), which indicated good stability of the formulation. The zeta potential values of F1, F2, and F3 were all reduced, with values of −3.42 ± 0.30, −4.15 ± 0.10, and −2.58 ± 0.30, respectively. However, the zeta potential values of F4, F10, F11, and F16 showed an increase of −20.03 ± 0.30, −19.03 ± 0.00, −19.03 ± 0.40, and −25.70 ± 0.30, respectively. The zeta potential was significantly influenced by the concentration of ethanol (B), the type of lipid (D), and the type of penetration enhancer (E). On the basis of their statistical interpretation, we can see how important each factor is in relation to other dependent variables. Keeping zeta potential values in the –25 to −35 mV range was our primary goal. As a result, the antagonistic effect (negative values) was preferred.

##### Effect of the Ethanol Concentration on the Zeta Potential (ZP)

The penetration-enhancing properties of ethanol are well known [57]. When it comes to ethosomal systems, vesicles with Z-potential, stability, entrapment efficiency, and increased skin permeability benefit from ethanol. Ethanol concentrations in ethosomal systems have been reported to range from 10% to 50% [26,58]. The solvent properties and edge activation mechanism of ethanol result in some steric stability, which in turn modifies the net charge of the systems. These findings were in agreement with those of Garg et al. and Yang et al. [23,53]. Electrostatic repulsion prevents the vesicular system from becoming aggregated because of the negative charge provided by ethanol. The vesicular charge is shifted from positive to negative due to the high ethanol concentration in ethosomes [24,26]. In addition, ethanol was found to have stabilizing effects, according to reports by Verma et al. and Dubey et al. [33,59]. The lowest ethanol concentration (10%) was advantageous because ZP values were improved as a result of its concentration. The surface charge and ethosome stability were both improved when ZP values were in the range of −19.03 ± 0.40 to −25.73 ± 0.40 mv. However, as the ethanol concentration rose to 40%, the zeta potential decreased (−1.10 ± 1.40 to −4.15 ± 0.10) to low values (unfavorable). Ethanol concentration significantly enhanced vesicles’ “zeta potential” (*p* < 0.05) as ethanol concentrations were found to be between 10% and 25% (mid-concentration range). According to Ogiso et al. [60], the skin permeation properties of negatively charged vesicles were superior to positively charged ones. The ethanol (10–25%) content in these nanocarriers is primarily responsible for the interpretation of the negative charge of the zeta potential in ethosomal systems. The polar head groups of phospholipids are negatively charged by ethanol, resulting in an electrostatic repulsion. As a result, ethosomal vesicles would be less likely to aggregate, increasing the stability of these nanocarriers. Another study by Abdulbaqi et al. [61] found the same thing. As ethanol concentration increased to 40%, ZP decreased, possibly due to the solubilization of phospholipids in the ethanol.

##### Effect of the Lipid Type on the Zeta Potential (ZP)

The vesicles’ surface charge was increased, and their stability was improved when they contained (lecithin lipid type) −19.03 ± 0.40 mv to −25.73 ± 0.40 mv in the formulations. The vesicles’ zeta potential values ranged from −2.58 ± 0.30 to −3.42 ± 0.30 mv in formulations containing type of lipid (Phospholipone 90 G).

The above findings indicated that the particles in the suspension carried anionic charge; an increase in soya lecithin concentration increased the anionic charges in the system, and thus it resisted agglomeration [62]. For instance, Kateh Shamshiri et al. [63] found that the skin’s negative surface charge is favorable for stabilizing and improving transdermal penetration of human growth hormone as a consequence of electrostatic repulsion between these charge on the skin surface and lecithin soybean phospholipid nano-transfersomes. When lecithin was added to the mixture, it significantly improved the zeta potential values of the vesicles (*p* < 0.05). The zeta potential with Phospholipon 90G, on the other hand, was found to be extremely low (non-favorable).

##### Effect of the Penetration Enhancer Type on the Zeta Potential (ZP)

When propylene glycol was used as a penetration enhancer in the preparation of VLT-ETHs, the zeta potential values of the vesicles were clearly increased. When propylene glycol was used as a penetration enhancer, zeta potential values ranged from −20.03 ± 0.30 to −25.70 ± 0.30 in the formulations, and this could be noted. The zeta potential values of the vesicles ranged from −2.58 ± 0.30 to −4.15 ± 0.10 for formulations containing Transcutol P. Therefore, the stability of ethosomes was increased by increasing the surface charge. On the basis of these findings, the propylene-glycol-incorporated ethosomal formulation F16 showed the greatest stability. It was found that for penetration enhancer (propylene glycol/Transcutol P), a 10% *v*/*v* concentration of penetration enhancer interacted with PL’s bilayer and allowed the hydrocarbon chain to pass through, increasing the bilayer’s flexibility. Propylene glycol was used as a penetration enhancer in the optimization study, and its concentration was set at 10%. A variation in penetration enhancer type (propylene glycol/transcutol P) appears to affect the charge that vesicles carry, as shown by the zeta potential measurement study.

When the ethanol concentration decreased, the ZP values increased. Adding lecithin and propylene glycol to ethosomal formulations also improved their ZP values. Nevertheless, neither the lipid concentration nor the sonication time had any significant effect.

#### 3.1.3. Effect of the Independent Factors on the Entrapment Efficiency (***EE***%)

The effectiveness of VLT entrapment within the formulations F1-F16 was tested in order to investigate the impact of ethosome composition on the drug-loading capacity of ethosome formulation, specifically, the amount of ethanol and lipid. It was found that the EE of VLT into the ethosomal formulation ranged between 74.27 ± 0.02 and 96.38 ± 0.01%, which is shown in Table 2. Table 3 shows that the ANOVA results indicated that ethanol quantity had no significant impact on ***EE***%. Conversely, both the lipid concentration and the lipid type were significant factors in the response to entrapment efficiency. The ethosomes’ entrapment efficiency was shown to be negatively influenced by lipid concentration, as displayed in Table 2. The preferred lipid type was lecithin.

##### Effect of the Concentration of Lipid on the Entrapment Efficiency (***EE***%)

For example, as lipid concentrations dropped, the ethosomal formulations’ ***EE***% increased. There was a significant difference between the formulations containing concentrations of lipids 1% and 4% when it came to the ***EE***% of the vesicles, which was found to be between 92.97 ± 0.01 to 96.38 ± 0.01 and 83.51 ± 0.02 to 91.44 ± 0.01%.

However, the maximum ***EE***% of F4 (96.38 ± 0.01 %) was obtained from a lipid concentration of 1%, which indicated a good formulation’s ability to load the drug into it. Entrapment efficiency was significantly increased (*p* < 0.05) when the PL concentration in formulations was reduced.

##### Effect of the Type of Lipid on the Entrapment Efficiency (***EE***%)

For formulations containing the lipid type (lecithin), ethosomal vesicles were found to be entrapped with an efficiency of between 90.01 ± 0.01 and 96.38 ± 0.01%. For F-9’s Phospholipon 90 G-based 2, we found that formulations containing Phospholipone 90 G (PL 90G) lipids had the lowest entrapment efficiency value. Because of this, lecithin inclusion significantly increased entrapment efficiency (*p* < 0.05).

Thus, lecithin would be the lipid of choice for the optimization study’s subsequent formulations. As a result, as lipid concentration dropped, ***EE***% increased. Furthermore, the addition of lecithin to ethosomal formulations improved their ability to encapsulate the drug. Despite this, the ethanol concentration, sonication time, or penetration enhancer type had no significant effect.

#### 3.1.4. Effect of the Independent Factors on the Polydispersity Index (PDI)

Because nanoparticle size and polydispersity index (PDI) affect the safety, stability, efficacy, and in vivo performance of nanoparticles as drug delivery systems, these parameters are critical for nanoparticle characterization [48]. To measure homogeneity of the formulation, it is necessary to know the distribution of vesicles. The width of the size distribution was quantified using the polydispersity index (PI).

PDI values under 0.30 suggest that the formulations’ particle populations are homogenous, but those over 0.30 suggest that they are heterogeneous [64]. Mura et al. and Tefas et al. [65,66] reported that PDI values less than 0.5 can also be accepted. PDI values of 0.24–0.38 indicate a fairly uniform and homogeneous particle size distribution, as shown in Table 2 [67]. Polydispersity index of 0.38 ± 0.00 indicates a homogeneous population of vesicles, which may ultimately lead to a decrease in mean particle size 133.80 ± 4.10 nm [51].

##### Effect of the Lipid Concentration on Polydispersity Index (PDI)

Lipid concentrations decreased the PDI of ethosomal formulations. The PDI of the vesicles was found to be between 0.24 ± 0.20 and 0.45 ± 0.00 for formulations that contained concentrations of 4% lipid, while the range for formulations that contained 1% lipid was found to be between 0.64 ± 0.00 and 1.00± 0.00. It was found that when PL concentration was increased from 1–4 percent *w*/*v*, the polydispersity index values of formulation were significantly decreased (*p* < 0.05).

##### Effect of Ethanol Concentration on Polydispersity Index (PDI)

With increasing ethanol concentrations, the PDI of ethosomal formulations decreased. To compare, the PDI values of the vesicles were found to be in the range of 0.24 ± 0.20–0.45 ± 0.00 for formulations with a 40% concentration of ethanol, but the values for formulations with 10% concentration were in the range of 0.68 ± 0.20–1.00± 0.00. Consequently, it was discovered that increasing the concentrations of ethanol (10–40 percent *w*/*v*) significantly decreased (*p* < 0.05) formulation polydispersity index values. Accordingly, the PDI decreased with increasing lipid and ethanol concentrations. The sonication time, lipid type, or penetration enhancer type had no significant impact on the results either.

#### 3.1.5. Summary of the Screening Results

In the screening stage, it was possible to determine which parameters for VLT-ETH preparation had significant influences and thus required further optimization, as well as what factors’ levels needed to be adjusted for significance. Lipid concentration and ethanol concentration in the VLT-ETHs formulations were the two parameters that were significant, were selected as independent factors, and were qualified for further optimization.

Because the sonication time was not statistically significant, the experiment can be retested with optimization. Lecithin was used to fix the lipid type, and propylene glycol was used to fix the penetration enhancer type. Because only small-sized particles can penetrate through the skin, and ethanol concentration significantly influences drug permeability, particle size and ethanol concentration influence skin permeability [44]. The VLT-ETH formulation parameters can be improved by using a lipid concentration of around 1% and an ethanol concentration level of around 25%. Particle size, zeta potential, polydispersity index, and entrapment efficiency were all measured.

### 3.2. Optimization Study Selection of the Most Significant Factors

From the screening for build information, we were able to identify the factors that had a significant impact on VLT-ETH preparation, as well as the levels that needed to be adjusted for those factors that were less significant. The optimum levels of the formulation parameters of VLT-ETHs were gained with the least number of experiments and in order to optimize the most significant factors obtained from the screening study. Optimized multi-formulations with optimum particle size, polydispersity index, Z-potential, and entrapment efficiency were created as result of this process.

A total of 18 runs with varying levels of three significant factors were conducted in random order in order to evaluate the most important factors. Table 4 shows the three factors that were used in the optimization study. A: ethanol concentration; B: lipid concentration, and C: sonication time. Dependent responses were particle size (PS) (Y1), polydispersity index (PDI) (Y2), zeta potential (ZP) (Y3), and entrapment efficiency (***EE***%) (Y4). It was found that a 25% concentration of ethanol and a 1% concentration of lipid were optimal for screening purposes. Lecithin is a good source of lipids. Propylene glycol should be the only type of penetration enhancer. Minimizing particle size (PS) (Y1) and polydispersity index (PDI) (Y2), achieving a zeta potential value (Y3) of −45 mV to −25 mV, and maximizing the entrapment efficiency (***EE***%) (Y4) were the objectives of the study. Results of the ANOVA are summarized in Table 5, including *p*-values. Items with a *p*-value of more than 0.05 were omitted from the study.

#### 3.2.1. Effect of the Independent Factors on Particle Size (PS)

Table 4 and Table 5 show that nano-sized ethosomes were developed. The PS ranged from 51.33 ± 0.60 nm of F9 to 253.27 ± 9.40 nm of F2.

A significant decrease in particle size (favorable) was observed (*p* < 0.05) when the lipid concentrations were in the range of the mid-concentration range (2.5 percent *w*/*v*), as shown in Table 5. It was discovered that the vesicles ranged in size from 51.68 ± 0.90 nm to 156.50 ± 3.20. Lipid concentrations of around 2.5% produced the smallest particles. This means that the lipid concentration would be 2.5% of the optimized formulation preparation.

#### 3.2.2. Effect of the Independent Factors on Polydispersity Index (PDI)

Table 5 shows that all of the optimization factors studied had a *p*-value of more than 0.05, so they were removed from the table.

#### 3.2.3. Effect of the Independent Factors on Zeta Potential (ZP)

In addition to skin–vesicle interactions, ethosomal vesicle charge had a significant impact on vesicular properties such as stability and vesicle–vesicle interactions. Additionally, dispersed systems require ZP of −35 to −25 mV for electrostatic stabilization [56]. Table 4 and Table 5 show that nano-sized VLT ethosomes were prepared. ZP values ranged from −40.70 ± 0.80 mv of F6 and −61.13 ± 4.10 mv of F4. It was found that formulation F-4 had the lowest zeta potential and the best stability, with a value of (−61.13 ± 4.10) mv. The ZP of the vesicles was found to be in the range of −40.70 ± 0.80 to −49.53 ± 1.10 mv in the formulations containing lipid concentrations of 1–1.75%. The ZP of the vesicles ranged from −54.67 ± 2.00 mv to −62.13 ± 4.10 mv in the formulations containing 2.5% lipid.

The maximum (enhanced) zeta potential value of −62.13 ± 4.10 mv was obtained with the preparation of F-4 with around 2.5% of lipid concentration. ZP values were enhanced (decreased) by using lecithin as a lipophilic lipid. This means that the lecithin concentration in the optimized formulation would be 2.5%. Zeta potential was significantly reduced by 2.5% lecithin concentration.

#### 3.2.4. Effect of the Independent Factors on Entrapment Efficiency (***EE***%)

There was an ***EE***% of VLT into the ethosomal formulation that was performed, and it was found to be between 87.30 ± 2.1 and 96.90 ± 0.3, as illustrated in Table 4. The ANOVA results, which are shown in Table 5, suggest that the ethanol and lipid concentrations had no effect on the ***EE***%. Conversely, sonication time was a significant factor in entrapment efficiency. Table 5 shows that ethosome entrapment efficiency was negatively affected by sonication time, as indicated by a negative sign. The ***EE***% of the vesicles was found to range from 92.27 ± 0.90 to 96.90 ± 0.30% in the formulations prepared at sonication time (5.50 min). To determine the EE percentage of the vesicles, we used formulations that had been sonicated for 7 min, and the results showed that the range was between 87.30 ± 2.1 and 89.87 ± 0.3%.

A maximized entrapment efficiency of F16 was 96.90 ± 0.30%, achieved with an ultrasonication time of 5.50 min. Conversely, it was found that 87.30 ± 2.1% of F5’s efficiency was reduced, prepared at a 7.51 min sonication time.

Subsequently, we can estimate that the optimized formulation’s sonication time would be 5.50 min. A sonication time of 5.50 min had a significant impact on the encapsulation efficiency.

### 3.3. Selection of the Optimized VLT-ETHs Formulation

There were three goals: to minimize PS and PDI in the range of 0.1–0.3, to maximize ZP in the range of −45 to −25 mV, and to maximize the percentage of EE. The ideal ethanol concentration, lecithin concentration, and sonication time for the VLT-ETH formulation were 24.47% *v*/*v*, 2.5% *w*/*v*, and 5.50 min, respectively.

Figure 1a shows that the vesicular size of the optimized VLT-ETH formulation was 45.8 ± 0.5 nm in diameter, with a PDI of 0.32 ± 0.02, which is shown in Table 6. The particle size of the optimized VLT-ETH formulation was best suited for topical application, and the PDI value indicated that there was a homogeneous distribution of a particle monodisperse population within the formulation. Since the stratum corneum tends to hold on to vesicles larger than 600 nm, the skin’s lipid layer may form as they dry [48,49]. Vesicles smaller than 300 nm were able to deliver their payload to some extent. Particles smaller than 70 nm were required for maximum drug delivery to deeper skin layers [46]. Additionally, the PDI values less than or equal to 0.3 indicated a homogeneous distribution of particles in a monodisperse population within the formulation, whereas values greater than or equal to 0.3 indicated a heterogeneous distribution.

In addition, as shown in Figure 1b, its zeta potential was measured, and the result is shown in Table 6 as being −51.4 ± 6.3 mV. The inclusion of ethanol acted to shift the surface charge and make the surface charge negative. As a result of the electrostatic repulsion, the formulation’s negative charge may prevent vesicle aggregation, which in turn would improve ethosomal stability [68]. The ethosomal formulation’s VLT entrapment efficiency was found to be 94.24 ± 0.2%. Since 2.5% lecithin concentration and lecithin lipid type resulted in an increase in the percentage of ***EE***%, it can be concluded that the ***EE***% of ethosomes was positively affected. Entrapment efficiency of the ethosomal formulation was found to be 94.24 ± 0.22%. As the ***EE***% of lecithin ethosomes increased, it can be concluded that 2.5% lecithin concentration and lecithin lipid type had a positive effect.

### 3.4. Transmission Electron Microscopy (TEM)

A 100× optical microscope magnification was used to verify the existence of vesicular structure in the prepared ethosomal formulations. It was observed that ethosomal vesicles are multilamellar in structure and of a very uniform shape and size. A spherical structure was observed. The vesicle population was dense, but aggregates were rare. TEM was used in order to additionally investigate the optimized VLT-ETHs’ shape and size distribution. As illustrated in Figure 2, TEM photomicrographs of the optimized VLT-ETHs indicated the presence of vesicles with a spherical structure and a smooth surface. Although the TEM image of vesicles did not reveal the lamella, it confirmed the vesicles’ size uniformity and symmetry.

A possible explanation for the results of the vesicle structure examination might be attributed to the influence of ethanol on PL, which assisted in providing a thinning effect on the outer membrane of the vesicles [23]. An explanation for the multilamellar structure of ethosomes could be that ethanol improved the flexibility and fluidity of PL bilayers, which allowed them to self-assemble [1].

### 3.5. Differential Scanning Calorimetry (DSC)

With the help of the DSC instrument, thermal behavior and various interactions between the drug (valsartan), physical mixture (valsartan and soya lecithin), and optimized formulation (VLT-ETHs) were effectively examined. The results of DSC experiments showing endothermic peaks of drug, physical mixtures (drug + lecithin), and optimized VLT-ETHs are depicted in Figure 3. The disappearance or shifting of endothermic or exothermic peaks indicated that the crystalline structure of the drug changed.

The DSC thermogram of valsartan exhibited a sharp and high intensity peak at 100.29 °C with an enthalpy of −22.20 J/g, and this was the endothermic peak observed for the drug. The presence of such a high endothermic peak demonstrated that the drug used was in a pure crystalline state [69]. Endothermic peaks at 236 °C were observed in the lecithin DSC thermogram. The drug peak in their physical mixture showed a shift with a decreased height. Lipid components melting and interacting with Valsartan could have been the cause of this. It was likely possible that the lipophilic drug could have been partially incorporated into the melted lipid. Kumari and Pathak, and Shaji and Lal had similar findings [33,70].

Thermal profiles for the optimized valsartan ethosomal formulation (VLT-ETHs) interestingly displayed endotherms at 42.95 and 141.51 °C for the PL. The melting point for the optimized formulation showed a peak of high intensity compared to the pure drug, suggesting that a more defective less crystalline morphology of the drug was achieved by the formulation (Figure 3) [71]. This infers that the optimized formulation VLT-ETHs showed no interaction with other excipients. An amorphous complex of valsartan was likely formed due to the lack of the drug’s melting endotherm. Valsartan’s interaction with the lipid bilayer structure was also proposed, explaining why the drug was more effectively encapsulated in the ethosomal formulation. Significant valsartan interaction with ethosomal constituents was highlighted by DSC and FTIR studies. According to Hathout et al. and Mohamed et al., these findings were consistent [72,73].

### 3.6. Fourier Transform Infrared Spectroscopy (FTIR)

For the detection of possible interactions and changes in the molecular order between the valsartan and the excipients used, FTIR spectroscopy was used as an analytical tool. It was possible to see evidence of molecular interactions in the form of new peak appearances, as well as changes in the prominent peak’s width or disappearance. The FTIR spectra of valsartan, the VLT-ETH formulation, and the physical mixture are illustrated in Figure 4. The FTIR spectrum of pure VLT showed well-defined characteristic absorption bands at 2952, 2869 cm^−1^ (stretching vibrations of aliphatic C-H), 1722 cm^−1^ (stretching vibrations of carboxylic C=O), 1593 cm^−1^ (stretching vibrations of amidic C=O), 1460 cm^−1^ (stretching vibrations of aromatic C=C), 1268 cm^−1^ (stretching vibrations of C-O), and 756 cm^−1^ (out-of-plane bending vibration of aromatic C-H), which have been studied extensively [71,74]. The optimized VLT-ETH spectrum demonstrated that there were no significant band shifts in the FTIR spectrum shown in Figure 4; however, some of the bands appeared to be widened due to the lower degree of crystallinity indicated by DSC results. It also showed a notable broadening in the O-H stretching at 3399 cm^−1^ (hydrogen-bonded), 2972, 2928 cm^−1^ (stretching vibrations of aliphatic C-H) of the drug and lipid, 1458 cm^−1^ (stretching vibrations of aromatic C=C) of the drug, and 1043 cm^−1^ (strong vibration of C-O) of the lipid.

The C=O stretching of valsartan disappeared from the formation of hydrogen bonds in the optimized VLT-ETH spectrum. It proposed a good indication of encapsulation of valsartan into the ethosomal vesicles by the shift in stretching vibrations of the drug’s amidic C=O (to 1640 cm^−1^), as well as the shift in stretching vibrations of C-O (to 1380 cm^−1^).

The hydrophobic interactions between valsartan and ethosomal ingredients are thought to be the cause of these changes because they promote the formation of stable vesicles with a good vesicle shape. Both Bodade et al. and Fathalla et al. found similar results [75,76].

### 3.7. In Vitro Drug Release Studies

In vitro drug release tests provide significant insights into a product’s in vivo behavior. This method determines the capacity of ethosomal vesicles to enhance drug permeation and the amount of drug exposed to absorption [59]. Transfer of drug molecules entrapped the ethosomes’ vesicles into the surrounding aqueous medium and caused subsequent diffusion into the receptor media via cellophane membrane-controlled drug diffusion from vesicles [77].

Figure 5 illustrates the cumulative VLT in vitro release profile from the optimized VLT-ETH formulation in phosphate buffer (pH 7.4). The optimized VLT-ETH formulation resulted in a significant increase in VLT in vitro release. Within 48 h, complete VLT release from the optimized formulation was observed.

Furthermore, there was no evidence of burst VLT release from the improved VLT-ETH formulation. The release of valsartan was regulated when the amount of PL was increased from 1% to 3% *w*/*v*. This could have been because the bilayer deformability of ethosomal vesicles resulted in lower in drug release, hence creating a barrier to AT diffusion.

Valsartan’s in vitro release can be influenced by variations in PL and ethanol concentrations. Additionally, when the ethanol content was increased to 27.5% *v*/*v*, drug release rose significantly. This could have been because the drug is more soluble in the hydroethanolic core, or because the ethosomal vesicles are more fluid. These findings corroborated those of Chourasia et al., Mohammed et al., and Iizhar et al. [55,73,77]. Due to the synergistic impact of propylene glycol and ethanol on the bilayers of vesicles, the presence of propylene glycol as a penetration enhancer increased the permeability of vesicles across the cellophane membrane. These findings corroborated those of Caddeo et al. and Zhao et al. [78,79].

In vitro release data were analyzed in order to select the model that best described the pattern of drug release, and the release profile data were subjected to a variety of different release models. The best fit was determined by the correlation coefficient (R^2^) value with the greatest yield. The Higuchi diffusion model (R^2^ = 0.9061) was determined to be the best fit for the studied VLT optimized formulation in phosphate buffer (pH 7.4). R^2^ values for the other kinetics release models were as follows: zero-order kinetics (R^2^ = 0.7919); first-order kinetics (R^2^ = 0.7926); second-order kinetics (R^2^ = 0.7932); Hixso–Crowell model (R^2^ = 0.7924); and Baker and Lonsdal equation (R^2^ = 0.8913). It is probable that the ethosomal vesicular system served as a reservoir for the continuous release of the entrapped drug.

### 3.8. Ex Vivo Nature of the Valsartan Ethosomal Gel (VLT-Ethogel)

Figure 6 represents the ex vivo permeation curves of valsartan from the valsartan ethosomal gel formulation (VLT-Ethogel). VLT permeability was seen to occur in two separate stages. The initial phase indicated a high rate of drug permeation that lasted around 8 h due to drug desorption from the ethosomes’ surface, followed by a slowing phase lasting at least 72 h due to drug diffusion within the ethosomes’ lipid bilayer. The proportions of VLT released from the VLT ethosomal-gel-based formulation (VLT-Ethogel) were 69.97 ± 3.44, 82.59 ± 2.88, 90.31 ± 1.01, and 96.94 ± 0.44%, respectively, after 12, 24, 48, and 72 h.

Ex vivo permeation tests were performed on the designed VLT-Ethogel formulation to determine the transdermal flow, which ranged from 10.4 ± 3.34 µg/cm^2^/h at 30 min to 232.66 ± 0.44 µg/cm^2^/h at 72 h. VLT release through the skin from an ethosomal-gel-based formulation (VLT-Ethogel) followed a Higuchi-diffusion release pattern, due to the fact that the Higuchi diffusion model had a higher correlation coefficient. Thus, Higuchi’s diffusion model best describes the permeation release of VLT from the formulated VLT-Ethogel formulations [30].

### 3.9. Pharmacokinetics Study of VLT in the Prepared Gel

#### Calculating the Pharmacokinetic Parameters

The pharmacokinetic variables and relative bioavailability of the tested VLT ethosomal gel (VLT-Ethogel) were studied in comparison to the oral solution of VLT. In order to conduct this experiment, this study was employed on rats. The gel and oral VLT solution were administered to eight rats. From each rat, plasma was collected, and the VLT concentrations in the plasma were measured. The VLT ethosomal-gel-treated group presented normal skin with well-defined epidermal and dermal layers and no indications of negative consequences (erythema, irritation, or inflammation). Additionally, the group’s horny layer look and thickness did not differ from that of the control samples. These results support the ethosomal system’s safety and tolerability.

The mean plasma concentration–time curve for the tested VLT ethosomal gel was estimated by comparing it to the mean plasma concentration–time curve for the VLT oral solution. All data are expressed as mean ± standard deviation, including pharmacokinetic parameters and relative bioavailability. Table 7 shows the mean plasma concentration of VLT (µg/mL) over time in hours following the application of ethosomal gel (VLT-Ethogel) to eight rats. To obtain the mean plasma VAL concentration–time graph shown in Figure 7, the corresponding pharmacokinetic parameters for oral VAL suspension and transdermal application of an optimized VAL nanoethosomal gel were used.

Oral VAL suspension and VAL nanoethosomal gel Cmax concentrations were 0.86 ± 2.14 μg/mL and 4.94 ± 8.02 μg/mL, respectively, at 2 and 16 h after administration. The AUC (0–∞) with oral treatment was 7.0 ± 2.94 (μg.h/mL), but the AUC (0–∞) with transdermal application of the VAL nanoethosomal gel was 137.2 ± 49.88 (μg.h/mL), providing the sustained release pattern of VLT from the tested ethosomal gel. The greater t_max_ value for transdermal VAL application compared to oral VAL administration could be attributed to the SC barrier feature, which retards VAL permeability [80]. In comparison, the suspension given orally appeared to have had a rapid release. Additionally, the mean value of AUC (0–∞) was larger for nanoethosomal VAL gel than for oral VAL suspension. Due to transdermal VAL application, the drug avoided first-pass hepatic metabolism, which could explain the increased bioavailability of VAL nanoethosomes. Additionally, the nanoethosomal gel showed an overall increase in biological half-life from 12.02 ± 0.45 h to 2.038 ± 0.50 h, compared to 2.038 ± 0.50 h with orally administered medication, which was attributed to the nanoethosomes’ sustained release ability [81]. Transdermal VAL administration resulted in a longer half-life because of the longer absorption time. Because of this, the VLT ethosomal gel designed for sustained drug delivery was successfully formed.

Ethosomes may act as both a carrier and a vehicle for controlled release, which would account for the efficient sustained release effect. In this study, it was found that incorporating VLT ethosomes into gel improved the bioavailability of VLT when compared to the oral solution. An increase in bioavailability could be attributed to the ethosomal formulation’s lipophilic nature. A high bioavailability VLT gel with a significant sustained release action can be made using VLT ethosomes, on the basis of the mean pharmacokinetic characteristics that were obtained.

## 4. Conclusions

The “hot method” and subsequent ultrasonication successfully improved the VLT-ETH formulations. The screening study was efficient for the initial screening phase. After taking into account all the variables and the formulations’ outcomes, it was possible to use lecithin as a chosen vesicle former and propylene glycol as a penetration enhancer. The inclusion of lecithin was found to improve the ZP and PS levels. To enhance the formulation characteristics of VLT-ETH, median lipid (2.5%) and ethanol (25%) concentrations were utilized. It was found that an ethanol concentration of 24.47% (*v*/*v*), a lecithin concentration of 2.5% (by *w*/*w*), and an ultrasonication period of 5.50 min were the best conditions for producing VLT-ETHs.

## Figures and Tables

**Figure 1 pharmaceutics-14-02268-f001:**
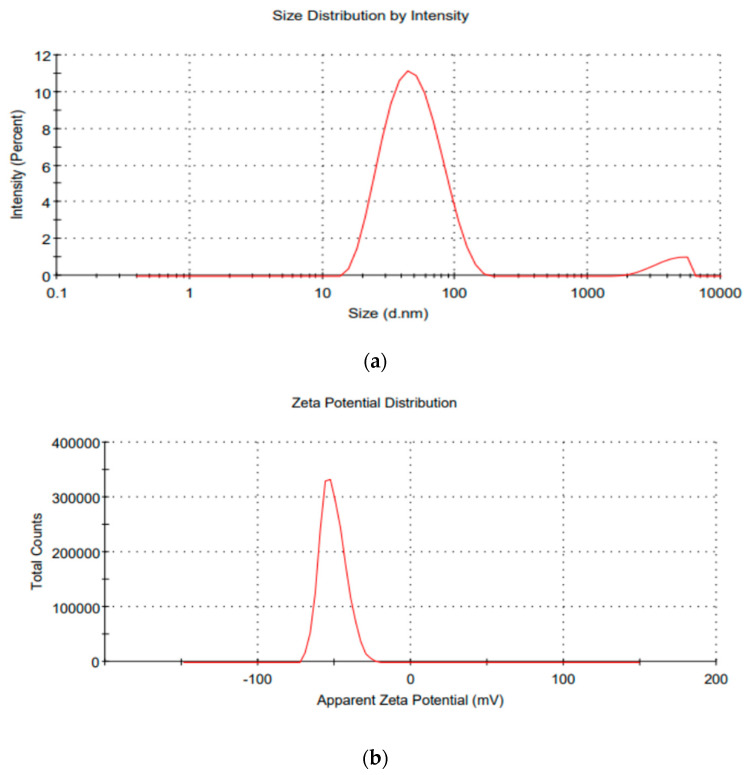
(**a**) Graphical view of the size, PDI properties. (**b**) Zeta potential of optimized VLT-ETH formulation. PDI, polydispersity index; VLT, valsartan; ETHs, ethosomes.

**Figure 2 pharmaceutics-14-02268-f002:**
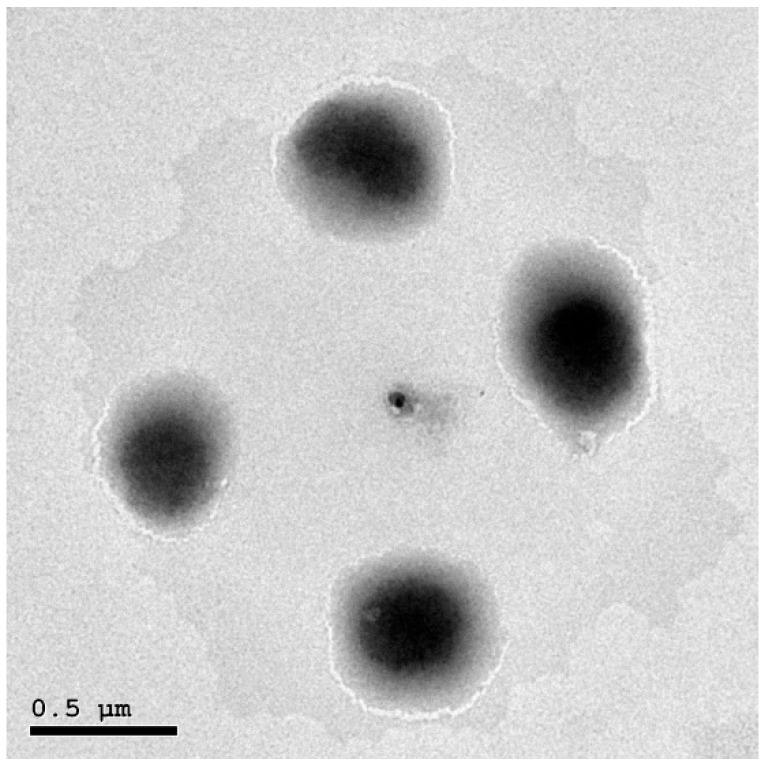
TEM photomicrograph of the optimized valsartan-loaded ethosome formulation.

**Figure 3 pharmaceutics-14-02268-f003:**
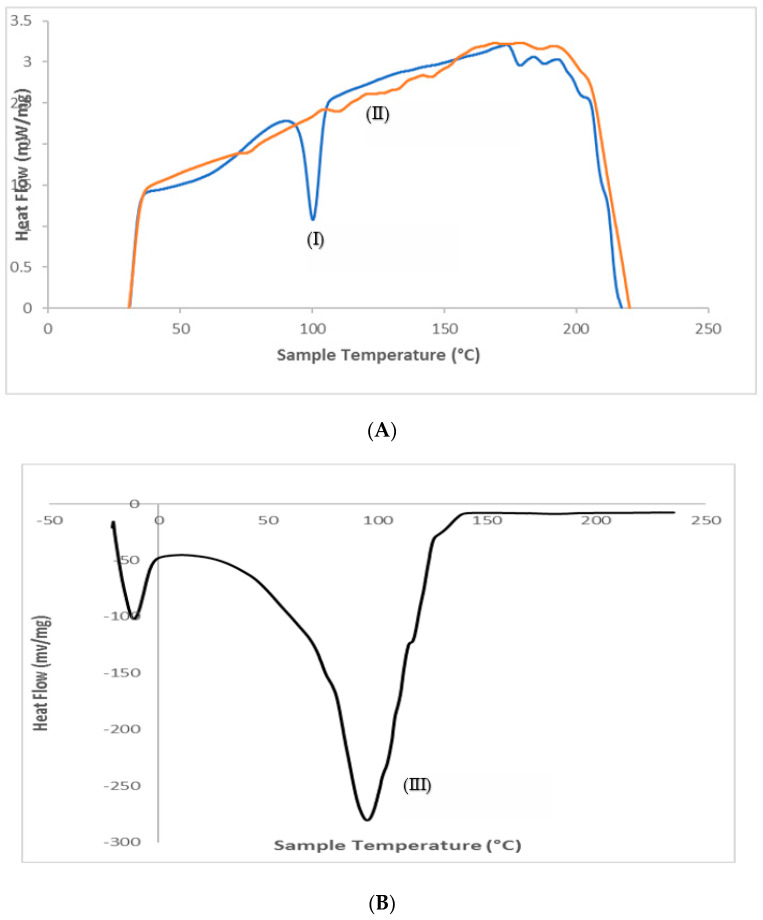
DSC thermograms of (**A**) (I) pure valsartan powder, (II) physical mixture, and (**B**) (III) optimized VLT-ETH formulation.

**Figure 4 pharmaceutics-14-02268-f004:**
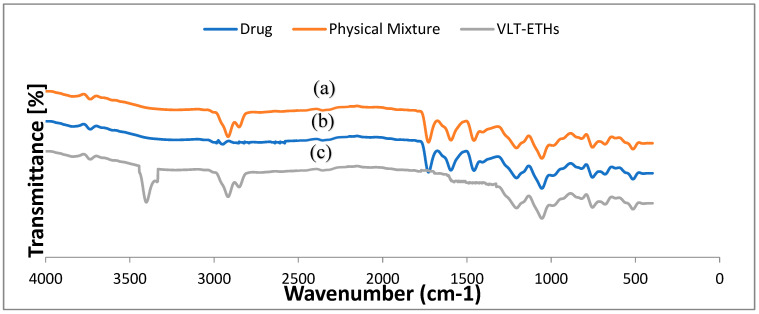
FTIR spectra of (a) physical mixture, (b) pure valsartan powder, and (c) optimized VLT-ETH formulation.

**Figure 5 pharmaceutics-14-02268-f005:**
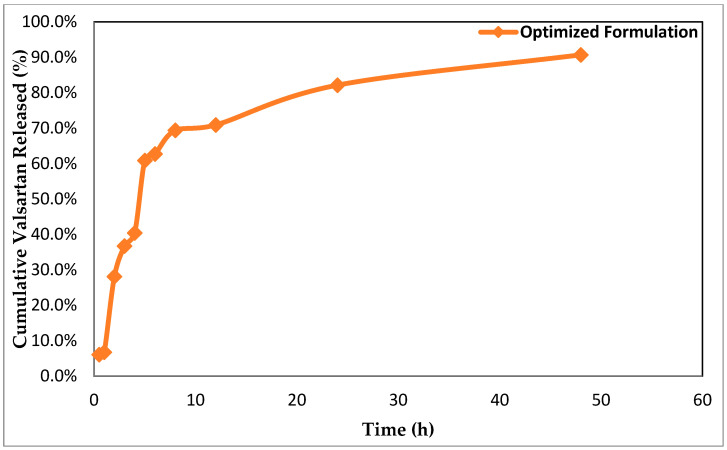
In vitro release of VLT from ethosome formulation (optimized formula).

**Figure 6 pharmaceutics-14-02268-f006:**
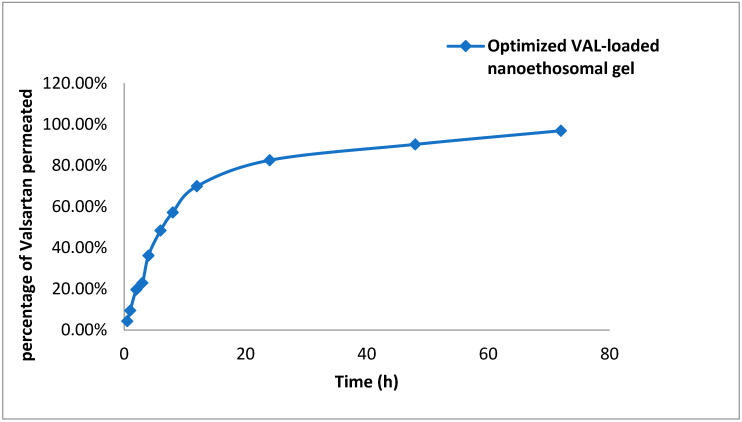
Ex vivo graph of valsartan ethosomal gel (mean ± S.D, *n* = 3).

**Figure 7 pharmaceutics-14-02268-f007:**
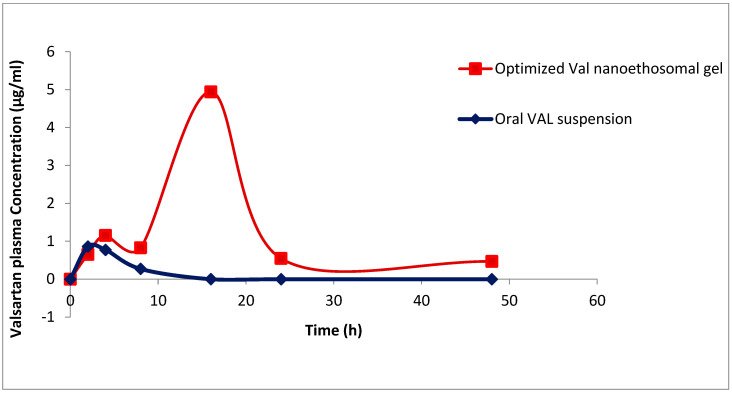
Plasma concentration time profiles of valsartan in rats after administration of oral suspension and optimized VAL nanoethosomal gel (mean ± SD).

**Table 1 pharmaceutics-14-02268-t001:** Build information for the initial screening study.

	Factor 1	Factor 2	Factor 3	Factor 4	Factor 5
Formulation	A: Lipid Concentration	B: Ethanol Concentration	C: Sonication Time	D: Lipid Type	E: Penetration Enhancer Type (10%)
	% *w*/*v*	% *w*/*v*	min		(%)
1	4	10	5	Phospholipone	Transcutol
2	4	40	5	Lecithin	Transcutol
3	4	10	5	Phospholipone	Transcutol
5	1	10	5	Lecithin	Propylene glycol
6	4	40	10	Lecithin	Propylene glycol
7	2.5	25	7.5	Lecithin	Transcutol
8	1	10	10	Lecithin	Transcutol
9	1	40	5	Lecithin	Transcutol
10	1.75	32.5	6.25	Phospholipone	Propylene glycol
11	4	10	10	Lecithin	Transcutol
11	4	10	10	Lecithin	Transcutol
12	1	40	10	Phospholipone	Propylene glycol
13	4	40	10	Lecithin	Propylene glycol
14	1	40	10	Phospholipone	Propylene glycol
15	1	10	10	Lecithin	Transcutol
16	1	10	5	Lecithin	Propylene glycol
Drug (mg)	80 in all formulations				
Water (%*v*/*v*)	To 100 in all formulations				

**Table 2 pharmaceutics-14-02268-t002:** The screening formulation parameters for the 16 formulations, with five factors, multiple levels, and measured responses.

F	A	B	C	D	E	R1	R2	R3	R4
1	4	10	5	PL 90G	Trans P	177.00 ± 3.20	−3.42 ± 0.30	91.44 ± 0.01	0.62 ± 0.00
2	4	40	5	Lecithin	Trans P	1723.00 ± 18.00	−4.15 ± 0.10	90.01 ± 0.01	0.24 ± 0.20
3	4	10	5	PL 90G	Trans P	389.57 ± 4.50	−2.58 ± 0.30	94.77 ± 0.02	0.68 ± 0.20
4	1	10	5	Lecithin	Prop. glycol	390.30 ± 3.10	−20.03 ± 0.30	96.38 ± 0.01	1.00 ± 0.10
5	4	40	10	Lecithin	Prop. glycol	133.80 ± 4.10	−1.10 ± 1.40	83.51 ± 0.02	0.38 ± 0.00
6	2.5	25	7.5	Lecithin	Trans P	369.73 ± 0.80	−11.13 ± 0.10	88.69 ± 0.02	0.47 ± 0.00
7	1	10	10	Lecithin	Trans P	313.83 ± 1.50	−25.73 ± 0.40	95.03 ± 0.01	1.00 ± 0.00
8	1	40	5	Lecithin	Trans P	635.83 ± 3.00	−7.43 ± 0.30	95.19 ± 0.01	0.41 ± 0.00
9	1.75	32.5	6.25	PL 90G	Prop. glycol	671.20 ± 2.80	0.44 ± 0.10	74.27 ± 0.02	0.98 ± 0.10
10	4	10	10	Lecithin	Trans P	1855.33 ± 4.00	−19.03 ± 0.00	84.48 ± 0.02	0.60 ± 0.00
11	4	10	10	Lecithin	Trans P	1855.33 ± 72.60	−19.03 ± 0.40	84.48 ± 0.01	0.60 ± 0.10
12	1	40	10	PL 90G	Prop. glycol	873.87 ± 10.80	0.18 ± 0.20	91.53 ± 0.02	1.00 ± 0.00
13	4	40	10	Lecithin	Prop. glycol	223.17 ± 5.20	−6.15 ± 0.20	95.90 ± 0.01	0.45 ± 0.00
14	1	40	10	PL 90G	Prop. glycol	1079.67 ± 46.30	0.68 ± 0.00	92.18 ± 0.01	0.64 ± 0.00
15	1	10	10	Lecithin	Trans P	435.03 ± 3.50	−11.80 ± 0.50	86.93 ± 0.01	0.43 ± 0.00
16	1	10	5	Lecithin	Prop. glycol	187.37 ± 6.30	−25.70 ± 0.30	92.97 ± 0.01	1.00 ± 0.00

Note: PL 90G: Phospholipon^®^ 90 G; Trans P: transcutol P; Prop. glycol: propylene glycol; F: formulation number; A: concentration of lipid (% *w*/*v*); B: concentration of ethanol (% *v*/*v*); C: sonication time (min); D: type of lipid; E: type of penetration enhancer; R1: particle size (nm); R2: zeta potential (mv); R3: entrapment efficiency (%); and R4: polydispersity index. All the formulations contained 80 mg of valsartan, 10% penetration enhancer (both types), and water up to 100% *v*/*v*.

**Table 3 pharmaceutics-14-02268-t003:** Coefficients and ANOVA results for the effects of the studied screening factors.

	Intercept	A	B	C	D	E
R1:PS	16.9112	11.5183			−12.436	
*p*-values		0.0004			0.0003	
R2:ZP	9.89743		−5.2385		−6.4237	3.50959
*p*-values			0.0094		0.0033	0.0482
R3:EE	94.3599	−4.3202			3.9952	
*p*-values		0.0010			0.0022	
R4:PDI	0.59507	−0.1425	−0.2017			
*p*-values		0.0029	0.0005			

Note: A: concentration of lipid (% *w*/*v*); B: concentration of ethanol (% *v*/*v*); C: sonication time (min); D: type of lipid; E: type of penetration enhancer; PS: particle size (nm); ZP: zeta potential (mv); EE: entrapment efficiency (%); and PDI: polydispersity index. The factor was considered insignificant when *p* > 0.05 and was removed from the table.

**Table 4 pharmaceutics-14-02268-t004:** Optimization study formulations of VLT-ETH formulations and the observed responses.

F	A	B	C	R1	R2	R3	R4
1	22.50	2.50	5.50	120.97 ± 17.90	0.39 ± 0.00	−41.33 ± 1.30	93.40 ± 0.00
2	22.50	1.00	7.00	253.27 ± 9.40	0.38 ± 0.00	−48.00 ± 2.90	91.56 ± 0.5
3	27.50	1.00	5.50	248.20 ± 9.10	0.36 ± 0.00	−47.30 ± 2.40	92.27 ± 0.9
4	27.50	2.50	5.50	101.60 ± 13.40	0.48 ± 0.10	−62.13 ± 4.10	91.44 ± 0.2
5	25.00	1.75	7.51	139.77 ± 15.80	0.48 ± 0.10	−49.27 ± 3.70	87.30 ± 2.1
6	25.00	0.49	6.25	120.87 ± 2.3	0.32 ± 0.00	−40.70 ± 0.80	90.10 ± 0.5
7	25.00	1.75	6.25	126.33 ± 0.50	0.28 ± 0.00	−52.73 ± 1.80	95.75 ± 0.5
8	29.20	1.75	6.25	214.70 ± 16.90	0.29 ± 0.00	−51.47 ± 3.00	93.89 ± 2.9
9	25.00	1.75	6.25	51.33 ± 0.60	0.24 ± 0.00	−49.53 ± 1.10	92.27 ± 0.4
10	22.50	2.50	7.00	51.68 ± 0.90	0.42 ± 0.10	−54.67 ± 2.00	93.65 ± 0.2
11	27.50	2.50	7.00	156.50 ± 3.20	0.27 ± 0.00	−58.07 ± 1.00	91.18 ± 0.2
12	20.80	1.75	6.25	202.90 ± 2.50	0.28 ± 0.00	−57.87 ± 1.20	94.14 ± 2.2
13	27.50	1.00	7.00	170.17 ± 6.20	0.34 ± 0.00	−58.20 ± 0.60	89.97 ± 0.3
14	25.00	1.75	6.25	123.27 ± 0.50	0.23 ± 0.00	−50.10 ± 1.30	92.69 ± 1.3
15	25.00	1.75	4.99	160.47 ± 2.80	0.26 ± 0.00	−51.40 ± 3.00	95.25 ± 1.4
16	22.50	1.00	5.50	59.39 ± 1.30	0.48 ± 0.10	−45.10 ± 5.30	96.90 ± 0.3
17	25.00	3.01	6.25	45.90 ± 0.40	0.32 ± 0.00	−58.70 ± 2.70	93.54 ± 2.1
18	25.00	1.75	6.25	81.87 ± 5.40	0.37 ± 0.00	−62.30 ± 3.20	91.09 ± 1.7

Note: F: formulation number; A: concentration of ethanol (% *v*/*v*); B: concentration of lipid (% *w*/*v*); C: sonication time (min); R1: particle size (nm); R2: polydispersity index; R3: zeta potential (mv); and R4: entrapment efficiency (%) (mean ± S.D, *n* = 3). All the formulations contain 80 mg of valsartan, 10% propylene glycol, and water up to 100% *v*/*v* (mean ± S.D, *n* = 3).

**Table 5 pharmaceutics-14-02268-t005:** Coefficient table for the effect of independent variables on dependent variables.

	Intercept	A	B	C	AÂ^2^
PS	107.15	15.45	−31.22		36.65
*p*-values		0.2878	0.0424		0.0202
ZP	−52.16		−3.51		
*p*-values			0.0417		
***EE***%	92.58			−1.54	
*p*-values				0.0087	

Note: A: concentration of ethanol, B: concentration of lipid, C: sonication time; PS: particle size, ZP: zeta potential, and ***EE***%: entrapment efficiency. The factor was considered insignificant when *p* > 0.05 and it was removed from the table.

**Table 6 pharmaceutics-14-02268-t006:** The actual values of the optimized VT-ETH formulation.

Number	Independent Variables			Dependent Variables Observed Values			
F	A	B	C	R1	R2	R3	R4
	%*v*/*v*	%*w*/*v*	minute	nm	mV	%	
1	24.47	2.50	5.50	45.8 ± 0.5	−51.4 ± 6.3	94.24 ± 0.2	0.32 ± 0.02

Note: F: formulation number; A: concentration of ethanol; B: concentration of lipid (% *w*/*v*); C: sonication time (min); R1: particle size (nm); R2: zeta potential (mv); R3: entrapment efficiency (%); and R4: polydispersity index (PDI) (mean ± S.D, *n* = 3). The optimized formulation contained 80 mg of valsartan, 10% propylene glycol, and water up to 100% *v*/*v*.

**Table 7 pharmaceutics-14-02268-t007:** Pharmacokinetics parameters of VAL-loaded nanoethosomal gel and oral VAL suspension (mean ± SD, *n* = 4).

	T_max_ (h)	C_max_ (µg/mL)	T_½_ (h)	AUC _(0–24)_ (µg.h/mL)	AUC _(0–∞)_ (µg.h/mL)
Optimized VAL-nanoethosomal gel	16	4.94 ± 8.02	12.02 ± 0.45	85.8 ± 37.33	137.2 ± 49.88
Oral VAL suspension	2	0.86 ± 2.14	2.038 ± 0.50	1.3 ± 12.62	7.0 ± 2.94

Abbreviations: Tmax: time to reach Cmax, Cmax: maximum serum concentration, AUC: the area under the serum concentration–time curve, SD: standard deviation.

## Data Availability

Not applicable.

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
