# Peer review of "Design, Formulation, and Characterization of Valsartan Nanoethosomes for Improving Their Bioavailability"

_pharmaceutics, 2022, doi:10.3390/pharmaceutics14112268_

Round 1

Reviewer 1 Report

I have been provided with the opportunity to review the article, entitled as, “Design, Formulation & Characterization of Valsartan Nano-2 ethosomes for Improving its Bioavailability”

Article is written well and con be considered further, but after addressing the following comments.

All the Italian word should be italic, “in-vivo

What is the melting point of used lipid, as authors have used temperature 40 °C?

Can author justify the 60rpm for sonication?

Is this enough speed to reduce the size of the vesicle to less than 100 nm size?

What was the limit of detection and limit of quantification of the drug and how they were determined?

In figure 2, the scale 0.5µm, does not confirm the claimed particle size of the ethosomes?

For compression, author can add the TEM image for unloaded ethosomes

Figure 3A, why there is a difference between two thermograms. Can physical mixture effect the thermal behaviour of the drug?

How author can explain the Figure3B?

Peaks in Figure 3A and 3B, should be labelled properly

IR scan in Figure 4 should be properly labelled

Is it physically applicable to maintain the gel on the skin for 48hours on the animal skin? Can author justify this study?

How author can justify that amount of permeated drug is greater than that of drug released?

Authors should add statistical analysis with suitable statistical test

Conclusion should be improved and results should not be repeated in the conclusion section.

Article should be revised comprehensively for grammatical and language corrections.

Reviewer 2 Report

page 3, line 104 please review the entire paragraph. it starts with"the organic phase was added to... at the end it seems to be cut off. 

authors are invited to describe briefly  the US patent  mentioned and state the differences with this paper. 

Round 2

Reviewer 1 Report

Some points are still need to be addressed 

 Is this enough speed to reduce the size of the vesicle to less than 100 nm size? Please provide suitable references.

What are the units of LOD and LOQ?

-       In figure 2, the scale 0.5µm, does not confirm the claimed particle size of the ethosomes? This point still needs to be addressed.
